# The Comparative Effects of Different Types of Oral Vitamin Supplements on Arterial Stiffness: A Network Meta-Analysis

**DOI:** 10.3390/nu14051009

**Published:** 2022-02-27

**Authors:** Alicia Saz-Lara, Iván Cavero-Redondo, Vicente Martínez-Vizcaíno, Isabel Antonia Martínez-Ortega, Blanca Notario-Pacheco, Carlos Pascual-Morena

**Affiliations:** 1Health and Social Research Center, Universidad de Castilla-La Mancha, 16171 Cuenca, Spain; alicia.delsaz@uclm.es (A.S.-L.); vicente.martinez@uclm.es (V.M.-V.); isabela.martinez@uclm.es (I.A.M.-O.); blanca.notario@uclm.es (B.N.-P.); carlos.pascual@uclm.es (C.P.-M.); 2Facultad de Ciencias de la Salud, Universidad Autónoma de Chile, Talca 3460000, Chile

**Keywords:** arterial stiffness, pulse wave velocity, oral vitamin supplementation, adults, network meta-analysis

## Abstract

Arterial stiffness, a significant prognostic factor of cardiovascular disease, may be affected by dietary factors. Research on the effects of oral vitamin supplements on arterial stiffness and/or endothelial function has produced controversial results. Therefore, the aim of this network meta-analysis was to comparatively assess the effect of different types of oral vitamin supplements on arterial stiffness in the adult population. We searched the PubMed, Embase, Cochrane Library, and Web of Science databases for randomized controlled trials from their inception to 30 September 2021. A network meta-analysis using a frequentist perspective was conducted to assess the effects of different types of oral vitamin supplements on arterial stiffness, as determined by pulse wave velocity. In total, 22 studies were included, with a total of 1318 participants in the intervention group and 1115 participants in the placebo group. The included studies were listed in an ad hoc table describing direct and indirect comparisons of the different types of vitamins. Our findings showed that, in both pairwise comparison and frequentist network meta-analysis, the different types of oral vitamin supplements did not show statistically significant effects on arterial stiffness. However, when oral vitamin supplementation was longer than 12 weeks, vitamin D3 showed a significant reduction in arterial stiffness, compared with the placebo (ES: −0.15; 95% CI: −0.30, −0.00; −60.0% m/s) and vitamin D2 (ES: −0.25; 95% CI: −0.48, −0.02, −52.0% m/s). In summary, our study confirms that oral vitamin D3 supplementation for more than 12 weeks could be an effective approach to reduce arterial stiffness and could be considered a useful approach to improve vascular health in patients at high risk of cardiovascular disease.

## 1. Introduction

Arterial stiffness is related to the onset of vascular aging [1] and could be considered a contributing factor for the development of cardiovascular disease (CVD) [2]. Inflammation and oxidative stress are possible mechanisms of arterial stiffness. Both mechanisms are able to induce changes in the endothelium and arterial wall structure through processes such as smooth muscle cell proliferation, collagen deposition, and elastin fragmentation [3]. Using disease risk stratification strategies to prevent the alteration of the vascular function and structure during the preclinical phases of disease may have different health benefits [4,5]. Pulse wave velocity (PWv) measurement is the gold standard method for the noninvasive assessment of arterial stiffness [2,6]. PWv is a useful surface indicator for determining cardiovascular risk [2,7,8] and targeting potential organ damage. However, the effects of reducing arterial stiffness, as measured by PWv, over time have yet to be determined [5].

Arterial “de-stiffening” is a topic of particular interest in preventing CVD. Currently, socioeconomic improvement and nutritional changes observed in the general population are associated with an increase in the prevalence of cardiometabolic diseases [9], with global trends of nutritional insufficiency [10]. Since diets are modifiable, their different components may increase or decrease the progression of CVD. Previous studies have shown that diets such as the Mediterranean diet that are rich in antioxidants and nutrients with anti-inflammatory properties can improve vascular function [11,12,13,14]. Furthermore, vitamin supplements are commonly used in Western countries, and observations show that, while vitamin supplements may provide health benefits, the potential health risks must also be considered [15].

Although there are several previously published systematic reviews and meta-analyses [16,17,18,19,20,21,22,23] on the effects of different types of oral vitamin supplements on arterial stiffness and/or endothelial function, and although the number of samples included in the assessment of some types of oral vitamin supplements is limited (and not all types of vitamins have been tested for arterial stiffness), none of them has quantitatively assessed the effects of different types of oral vitamin supplementation (folic acid or vitamin B9, ascorbic acid or vitamin C, calcitriol or vitamin D, ergocalciferol or vitamin D2, cholecalciferol or vitamin D3, tocotrienol or vitamin E) on arterial stiffness. Therefore, the aim of this network meta-analysis (NMA) was to assess the effects of different types of oral vitamin supplements on the reduction in arterial stiffness.

## 2. Materials and Methods

This NMA was reported in compliance with the preferred reporting items for systematic review incorporating NMA (PRISMA-NMA) guidelines [24] and the Cochrane Collaboration Handbook [25]. This study was registered in PROSPERO (Registration Number: CRD42021253233). This study included data from previously published studies, so it did not include individual patients and no prior ethical approval was required.

### 2.1. Search Strategy

Two reviewers (A.S.-L. and I.C.-R.) independently conducted systematic searches of the PubMed, Web of Science, Cochrane Library, and Scopus databases from their inception to 30 September 2021. Free terms in combination with Boolean operators were used to perform the search, following the PICO strategy (population, intervention/exposure, comparison, and outcome): “adults”, “young adults”, “older adults”, “elderly adults”, “adult population”, “adult subjects”, “vitamins”, “oral vitamin supplementation”, “vitamin B9”, “folic acid”, “vitamin C”, “ascorbic acid”, “vitamin D”, “calciferol”, “vitamin D3”, “cholecalciferol”, “vitamin D2”, “ergocalciferol”, “vitamin E”, “tocopherol”, “arterial stiffness”, “aortic stiffness”, “pulse wave velocity”, and “PWv”. Appendix A shows the search strategy. In addition, to achieve a more comprehensive search, references of included articles, and previous systematic reviews or meta-analyses in the field were reviewed.

### 2.2. Eligibility

Studies concerning the effects of oral vitamin supplements on arterial stiffness were included in the NMA. Studies were selected based on the following inclusion and exclusion criteria. Inclusion criteria: (1) studies: RCTs; (2) population: subjects older than 18 years old; (3) intervention: oral vitamin supplements (folic acid or vitamin B9, ascorbic acid or vitamin C, calcitriol or vitamin D, ergocalciferol or vitamin D2, cholecalciferol or vitamin D3, tocotrienol or vitamin E) with a duration of at least 4 weeks; (4) outcome: arterial stiffness measured by PWv. Exclusion criteria: (1) single-arm pre- and post-studies; (2) studies combining oral vitamin supplementation with pharmacological treatments; and (3) studies in which the type and dose of vitamins could not be estimated.

### 2.3. Study Selection and Data Extraction

Study selection and data extraction were performed independently by two researchers (A.S.-L. and I.C.-R.). The following information was extracted from the included studies: (1) reference (first author and year of publication); (2) country in which the study data were collected; (3) population characteristics (sample size (%female), mean age, type of population (healthy or with a specific disease)); (4) intervention characteristics (type of vitamins (folic acid or vitamin B9, ascorbic acid or vitamin C, calcitriol or vitamin D, ergocalciferol or vitamin D2, cholecalciferol or vitamin D3, tocotrienol or vitamin E), oral supplementation dose (frequency), length); (5) outcome: arterial stiffness measured by PWv (type of PWv (aortic PWv (a-PWv), brachial-to-ankle PWv (ba-PWv), brachial-to-radial PWv (br-PWv), carotid-to-femoral PWv (cf.-PWv), carotid-to-radial PWv (cr-PWv)), PWv device, baseline PWv levels, percentage of change) (Table 1).

### 2.4. Risk of Bias Assessment

Two researchers (A.S.-L. and I.C.-R.) independently performed an assessment of the methodological quality of the included RCTs using the Cochrane Collaboration tool for assessing the risk of bias (RoB2) [26] following the Cochrane manual for the systematic review of interventions [25] with information on authors, dates, and sources of each included manuscript blinded. This tool assesses the risk of bias based on six domains: selection bias, performance bias, detection bias, attrition bias, reporting bias, and other biases. Overall bias was rated “low risk of bias” when all domains were classified as “low risk”, “some concerns” when there was at least one domain classified as “some concern”, and “high risk of bias” when there was at least one domain classified as “high risk” or several domains classified as “some concerns”. Disagreements were handled by consensus or by a third reviewer (C.P.-M.).

### 2.5. Grading the Quality of Evidence

We used the Grading of Recommendations, Assessment, development, and Evaluation (GRADE) tool [27] to assess the quality of evidence and provide recommendations. Each outcome had a high, moderate, low, or very low evidence score, depending on study design, risk of bias, inconsistency, indirect evidence, imprecision, and publication bias.

### 2.6. Data Synthesis and Statistical Analysis

The included studies were qualitatively reported in an ad hoc table depicting direct and indirect comparisons of different types of oral vitamin supplements. Our NMA was conducted in accordance with the PRISMA-NMA statement [24] under a frequentist perspective by following these steps:-First, we used a network geometry graph to depict the trials in the network. In this graph, the size of the nodes was relative to the number of participants in trials receiving the intervention identified in the node, and the width of the solid line connecting the nodes was relative to the number of participants in trials directly comparing the two interventions. Dashed lines depict indirect comparisons between two interventions [28].-Second, the consistency assessment tested whether the intervention effect calculated from direct comparisons was robust with those calculated by indirect comparisons. For this purpose, we used the Wald test, and we evaluated local inconsistency using the side-splitting method.-Third, we performed a comparative assessment of the intervention effect by performing a standard pairwise meta-analysis for comparisons between interventions and placebo/other interventions. For this purpose, we used the DerSimonian–Laird random-effects method [29] to calculate a pooled effect size (ES) estimate and the respective 95% confidence intervals (CIs), and we estimated the pooled percentage change in m/s for oral vitamin supplement interventions. We examined statistical heterogeneity by calculating the I2 statistic, ranging from 0% to 100%. Depending on the I2values, heterogeneity was classified as unimportant (0% to 30%), moderate (30% to 50%), substantial (50% to 75%), or considerable (75% to 100%) [25]. In addition, we considered the corresponding p values. Finally, we calculated the statistic τ2 to establish the size and clinical relevance of heterogeneity. A τ2 estimate of 0.04 can be considered as low, 0.14 as moderate, and 0.40 as a substantial degree of the clinical relevance of heterogeneity [30]. We created both forest plots and a league table to depict these results.-Fourth, we calculated the effect of each intervention using NMA with a frequentist perspective [31]. Frequentist perspective draws a conclusion based on the level of statistical significance and the acceptance or rejection of a hypothesis.-Fifth, we used sensitivity, subgroup, and meta-regression analyses for the transitivity evaluation, and we verified that all study participants included in the NMA had, on average, a similar baseline effect distribution. We conducted a sensitivity analysis (systematic reanalysis while eliminating studies one at a time) to evaluate the strength of the summary estimates. We used subgroup analyses based on the mean population age (<65 years or >65 years), intervention length (<12 weeks or >12 weeks), vitamin type (water-soluble (vitamin B9 and vitamin C) or fat-soluble (vitamin D, vitamin D2, vitamin D3, and vitamin E)), and PWv type (central PWv (a-PWv and cf.-PWv) or peripheral PWv (ba-PWv, br-PWv, and cr-PWv)). We performed meta-regression analyses to address whether the mean age and intervention length, as continuous variables, modified the effect of oral vitamin supplementation interventions on PWv.-Sixth, once the effect size estimates of the efficacy of oral vitamin supplementation interventions were calculated, we ranked and plotted the interventions using rankograms [32]. In addition, we calculated the surface under the cumulative ranking (SUCRA) [28] for each intervention.-Finally, publication bias was assessed through visual inspection of the funnel plots, and Egger’s test [33].

We performed all analyses using Stata 15.0 (Stata, College Station, TX, USA).

**Table 1 nutrients-14-01009-t001:** Characteristics of the included studies.

Reference	Country	Population Characteristics			Intervention Characteristics			Outcome: Arterial Stiffness			
		Sample Size (% Female)	Mean Age (Years)	Type of Population	Type of Vitamins	Oral Supplement dose (Frequency)	Length (Weeks)	Type of PWv	PWv Device	Basal PWv Levels (m/s)	% Change (m/s)
Mangoni et al., 2002 [34]	United Kingdom	IG: 12 (66.7) CG: 12 (58.3)	IG: 39.7 ± 11.9 CG: 36.0 ± 12.6	Healthy	Folic acid (vit. B9)	5 mg (daily)	4	cf-PWv	Complior	IG: 8.4 ± 1.8 CG: 8.3 ± 1.1	IG: −7.1% CG: −6.0%
Mangoni et al., 2005 [35]	United Kingdom	IG: 13 (38.5) CG: 13 (53.8)	IG: 55. 3 ± 4.3 CG: 57.6 ± 4.7	DM2	Folic acid (vit. B9)	5 mg (daily)	4	cf-PWv	Complior	IG: 10.8 ± 2.5 CG: 10.9 ± 2.9	IG: −1.0% CG: 2.8%
Nightingale et al., 2003 [36]	United Kingdom	IG: 23 (26.1) CG: 15 (26.7)	IG: 57.8 ± 9.9 CG: 60.9 ± 6.3	CHF	Ascorbic acid (vit. C)	4 g (daily)	4	br-PWv	QVLP84	IG: 7.8 ± 1.9 CG: 7.6 ± 0.9	IG: 6.8% CG: 5.4%
Nightingale et al., 2007 [37]	United Kingdom	IG: 19 (15.8) CG: 18 (16.7)	IG: 64.0 ± 8.7 CG: 63.0 ± 8.5	CHF	Ascorbic acid (vit. C)	4 g (daily)	4	ba-PWv	QVLP84	IG: 9.8 ± 2.6 CG: 9.7 ± 3.8	IG: 92.0% CG: −27.0%
Dreyer et al., 2014 [38]	United Kingdom	IG: 20 (39.1) CG: 18 (26.3)	IG: 45.8 ± 10.0 CG: 48.8 ± 12.2	CKD	Ergocalciferol (vit. D2)	50,000 IU (weekly for one month) + 50,000 IU (monthly)	16	a-PWv	Vicorder	IG: 8.5 ± 1.1 CG: 8.5 ± 1.5	IG: −1.2% CG: 0.0%
Kovesdy et al., 2012 [39]	United States	IG1: 40 (0.0) IG2: 40 (2.0)	IG1: 67.6 ± 9.3 IG2: 69.3 ± 10.6	CKD	IG1: Ergocalciferol (vit. D2) IG2: Cholecalciferol (vit. D3)	IG1: 50,000 IU (single dose) IG2: 1 or 2 µg (daily)	16	a-PWv	Sphygmocor	IG1: 12.8 ± 3.5 IG2: 13.5 ± 3.9	IG1: 1.6% IG2: −1.5%
Forouhi et al., 2016 [40]	United Kingdom	IG1: 112 (43.8) IG2: 114 (43.0) CG: 114 (42.1)	IG1: 53.5 ± 8.7 IG2: 52.5 ± 8.2 CG: 52.4 ± 8.5	Pre-DM2	IG1: Ergocalciferol (vit. D2) IG2: Cholecalciferol (vit. D3)	IG1: 3300 IU (daily) IG2: 3300 IU (daily)	16	cf-PWv	Doppler MDII	IG1: 7.3 ± 2.7 IG2: 7.9 ± 2.0 CG: 7.4 ± 2.0	IG1: −2.3% IG2: −9.5% CG: 4.9%
Larsen et al., 2012 [41]	Denmark	IG: 55 (70.0) CG: 57 (68.0)	IG: 60.0 ± 12.0 CG: 61.9 ± 9.0	HT	Cholecalciferol (vit. D3)	3000 IU (daily)	20	cf-PWv	SphygmoCor	IG: 8.5 ± 2.3 CG: 8.7 ± 2.1	IG: 5.9% CG: 3.5%
Marckmann et al., 2012 [42]	Denmark	IG: 26 (26.9) CG: 26 (23.1)	IG: 71 (62–78) CG: 68 (59–76)	CKD	Cholecalciferol (vit. D3)	40,000 IU (weekly)	8	a-PWv	Millar SPT-301B	IG: 12. 0 (9.0–13.9) CG: 10.0 (7.8–13.2)	IG: 5.8% CG: −3.0%
Hewitt et al., 2013 [43]	Australia	IG: 30 (47) CG: 30 (57)	IG: 60 (53–71) CG: 67 (54–72)	CKD	Cholecalciferol (vit. D3)	50,000 IU (weekly for two months) + 50,000 IU (monthly)	24	cf-PWv	SphygmoCor	IG: 10.3 ± 4.0 CG: 10.3 ± 4.0	IG: −9.7% CG: 1.9%
Witham et al., 2013 [44]	United Kingdom	IG: 24 (100.0) CG: 25 (100.0)	IG: 41.7 ± 13.4 CG: 39.4 ± 11.8	Healthy	Cholecalciferol (vit. D3)	100,000 IU (single dose)	8	cr-PWv	SphygmoCor	IG: 8.0 ± 1.2 CG: 7.7 ± 1.7	IG: 6.3% CG: −3.9%
Mose et al., 2014 [45]	Denmark	IG: 25 (32.0) CG: 25 (40.0)	IG: 68.0 ± 9.0 CG: 67.0 ± 13.0	CKD	Cholecalciferol (vit. D3)	3000 IU (daily)	24	cf-PWv	SphygmoCor	IG: 9.7 ± 2.5 CG: 10.0 ± 2.0	IG: 8.2% CG: 1.0%
Pilz et al., 2015 [46]	Germany	IG: 100 (46.0) CG: 199 (48.0)	IG: 60.5 ± 10.9 CG: 59.7 ± 11.4	HT	Cholecalciferol (vit. D3)	2800 IU (daily)	8	NA	NA	IG: 8.4 ± 2.0 CG: 8.3 ± 2.1	IG: 1.0% CG: 4.1%
Witham et al., 2015 [47]	United Kingdom	IG: 25 (72.0) CG: 25 (80.0)	IG: 48.1 ± 12.0 CG: 50.7 ± 13.1	Chronic fatigue syndrome	Cholecalciferol (vit. D3)	100,000 IU (single dose)	24	cf-PWv	SphygmoCor	IG: 7.3 ± 2.6 CG: 8.3 ± 1.9	IG: −5.5% CG: −2.4%
Bressendorff et al., 2016 [48]	Denmark	IG: 22 (50) CG: 18 (34)	IG: 41.0 ± 9.1 CG: 44.5 ± 8.5	Healthy	Cholecalciferol (vit. D3)	3000 IU (daily)	16	cf-PWv	SphygmoCor	IG: 6.4 ± 1.4 CG: 6.7 ± 0.9	IG: 0.0% CG: −1.5%
Kumar et al., 2017 [49]	United Kingdom	IG: 58 (29.3) CG: 59 (32.2)	IG: 43.2 ± 11.8 CG: 45.2 ± 11.6	CKD	Cholecalciferol (vit. D3)	300,000 IU (two doses: baseline and 8weeks)	16	cf-PWv	SphygmoCor	IG: 8.0 ± 1.6 CG: 8.0 ± 1.7	IG: −11.8% CG: 3.8%
Sluyter et al., 2017 [50]	New Zealand	IG: 256 (40.0) CG: 261 (48.0)	IG: 64.5 ± 8.3 CG: 65.5 ± 8.8	HT, DM	Cholecalciferol (vit. D3)	200,000 IU (single dose) + 100,000 IU (monthly)	48	a-PWv	Mobil-O-Graph	IG: 9.3 ± 1.7 CG: 9.3 ± 1.7	IG: −1.1% CG: 0.0%
Gepner et al., 2012 [51]	United States	IG: 57 (100) CG: 57 (100)	IG: 64.1 ± 3.0 CG: 63.6 ± 3.1	Postmenopausal	Cholecalciferol (vit. D3)	2500 IU (daily)	16	cf-PWv	SphygmoCor	IG: 7.8 ± 0.9 CG: 8.0 ± 1.4	IG: -1.0% CG: 0.0%
Levin et al., 2017 [52]	Canada	IG1: 39 (28.0) IG2: 40 (30.0) CG: 40 (27.0)	IG1: 66.9 ± 11.7 IG2: 65.9 ± 15.3 CG: 64.5 ± 12.2	CKD	IG1: Calcitriol (vit. D) IG2: Cholecalciferol (vit. D3)	IG1: 0.5 µg (thrice weekly) IG2: 5000 IU (thrice weekly)	24	cf-PWv	SphygmoCor	IG1: 11.6 ± 3.8 IG2: 12.2 ± 4.2 CG: 10.7 ± 3.7	IG1: 5.2% IG2: 1.6% CG: −1.0%
Tomson et al., 2017 [53]	United Kingdom	IG1: 102 (49.0) IG2: 102 (50.0) CG: 101 (49.0)	IG1: 71.0 ± 6.0 IG2: 72.0 ± 6.0 CG: 72.0 ± 6.0	HT, heart disease, DM, stroke	Cholecalciferol (vit. D3)	IG1: 4000 IU (daily) IG2: 2000 IU (daily)	24	a-PWv	Arteriograph	IG1: 10.0 ± 1.9 IG2: 9.6 ± 1.6 CG: 9.7 ± 1.8	IG1: −2.0% IG2: 3.1% CG: 2.1%
Rasool et al., 2006 [54]	Malaysia	IG1: 9 (0.0) IG2: 9 (0.0) IG3: 9 (0.0) CG: 9 (0.0)	IG1: 21–30 IG2: 21–30 IG3: 21–30 CG: 21–30	Healthy	Tocotrienol (vit. E)	IG1: 80 mg (daily) IG2: 160 mg (daily) IG3: 320 mg (daily)	8	cf-PWv	SphygmoCor	IG1: 7.4 ± 0.7 IG2: 7.8 ± 0.7 IG3: 7.5 ± 0.6 CG: 7.8 ± 0.8	IG1: 1.3% IG2: −2.6% IG3: −4.0% CG: −1.3%
Stonehouse et al., 2016 [55]	Australia	IG: 28 (35.7) CG: 29 (37.9)	IG: 60.5 (56.5–65.8) CG: 61.0 (56.0–64.0)	DM2	Tocotrienol (vit. E)	420 mg (daily)	8	cf-PWv	Millar SPT-301	IG: 6.8 (5.9–7.6) CG: 7.2 (6.3–8.1)	IG: −7.0% CG: −13.3%

Data are shown as mean ± standard deviation (SD) or median (interquartile range); a-PWv: aortic pulse wave velocity; br-PWv: brachial-to-radial pulse wave velocity; cf-PWv: carotid-to-femoral pulse wave velocity; CG: control group; CHF: chronic heart failure; CKD: chronic kidney disease; cr-PWv: carotid-to-radial pulse wave velocity; DM: diabetes mellitus; HT: hypertension; IG: intervention group; NA: not available; PWv: pulse wave velocity; RCT: randomized controlled trial.

## 3. Results

### 3.1. Study Characteristics

A total of 22 studies [34,35,36,37,38,39,40,41,42,43,44,45,46,47,48,49,50,51,52,53,54,55] were included in this NMA (Figure 1). All studies were RCTs. The studies were published between 2002 and 2017 and were conducted in eight different countries, with the United Kingdom being the most frequently reported. The sample size of the studies ranged from 9 to 261 healthy or unhealthy adults (aged 21.0 to 72.0 years). The interventions ranged from 4 to 48 weeks, and the oral vitamin supplementation interventions included two studies for vitamin B9 171 [34,35], two studies for vitamin C [36,37], one study for vitamin D [52], three studies for vitamin D2 [38,39,40], fifteen studies for vitamin D3 [39,40,41,42,43,44,45,46,47,48,49,50,51,52,53], and two studies for vitamin E [54,55]. Regarding the type of PWv measured, different methods were included—namely, thirteen for cf.-PWv [34,35,40,41,43,45,47,48,49,51,52,54,55], five for a-PWv [38,39,42,50,53], one for ba-PWv [37], one for br-PWv [36], and one for cr-PWv [44]. The characteristics of the included studies are shown in Table 1.

### 3.2. Risk of Bias and GRADE

For the overall risk of bias 77.3% of the studies showed some concerns and 22.7% of the studies had a low risk of bias. With respect to specific domains, in the domain of randomization process, 86.4% of the studies were rated as low bias; in the domains of deviations from intended interventions, the missing outcome and the measuring of the outcome, 50.0%, 36.4%, and 36.4% of the studies, respectively, were rated as some concerns; finally, in the domain of reported results, 100.0% of the studies were rated as low risk of bias (Appendix A).

When the GRADE was evaluated, 75.0% of the pairwise comparison studies were rated as moderate and 25.0% as low (Appendix A).

### 3.3. Effect of Oral Vitamin Supplementation on Arterial Stiffness

Figure 2 displays the network geometry plot of the comparisons testing the effect of different oral vitamin supplementation interventions on arterial stiffness. Table 2 displays the ES estimates from direct studies separately (upper diagonal) from the indirect ES estimates (lower diagonal). In the pairwise analyses, although no significant results were shown, compared with the placebo group, all estimates were in favor of oral vitamin supplementation interventions, with the exceptions of vitamins C and E. In addition, oral vitamin D3 supplementation proved to be a better supplement for decreasing arterial stiffness than vitamin D2 supplementation (ES: −0.25; 95% CI: −0.48, −0.02; −36.0% m/s). In the frequentist NMA, no significant results were shown; however, oral vitamin supplementation interventions are in favor of reducing arterial stiffness, with the exception of vitamins C and E.

### 3.4. Probabilities

Oral supplementation with vitamin B9 was more likely to be the best (31.0%), and oral supplementation with vitamin D3 showed a higher SUCRA value (78.0%) (Appendix A).

Oral supplements with vitamin D3 versus the placebo group showed substantial heterogeneity (I2 = 63.4%, τ2 = 0.06). The other direct comparisons showed no heterogeneity (I2 = 0.0, τ2 = 0.00) (Appendix A).

### 3.5. Sensitivity Analysis, Subgroup Analyses, Meta-Regression Models, and Publication Bias

The estimate of the pooled ES was not significantly changed (either in magnitude or direction) when data from individual studies were eliminated from the analysis one study at a time.

Subgroup analyses based on mean age (<65 years or >65 years) showed that oral supplementation with vitamin D3 was effective in reducing arterial stiffness, compared with oral supplementation with vitamin D2 in adults <65 years (ES: −0.30; 95% CI: −0.57, −0.03; −58.0% m/s) (Appendix A). Subgroup analysis based on length of intervention (<12 weeks or >12 weeks) showed that oral supplementation with vitamin D3, compared with the placebo group, and oral supplementation with vitamin D3, compared with oral supplementation with vitamin D2, were effective in reducing arterial stiffness for interventions >12 weeks (ES: −0.15; 95% CI: −0.30, −0.00; −60.0% m/s; and ES: −0.25; 95% CI: −0.48, −0.02, −52.0% m/s, respectively) (Appendix A). For subgroup analyses based on the type of PWv (central or peripheral PWv), central PWv showed significant results in oral supplementation with vitamin D3, compared with oral supplementation with vitamin D2 (ES: −0.25; 95% CI: −0.48, −0.02; −36.0% m/s) (Appendix A). For subgroup analyses based on the type of vitamins (water soluble or fat soluble), no significant results were shown (Appendix A).

Meta-regression models showed that mean age and length of intervention intensity did not modify the effect of oral supplementation with vitamin D3 vs. placebo on arterial stiffness (Appendix A). Other comparisons were not performed because of an insufficient number of studies.

No publication bias was shown in the included comparisons (Appendix A).

## 4. Discussion

This NMA provides an overview of the evidence comparing different types of oral vitamin supplements on arterial stiffness. Our findings showed that even though overall, the different types of oral vitamin supplements showed no statistically significant effects, when oral vitamin supplementation was longer than 12 weeks, vitamin D3 showed a significant reduction in central arterial stiffness, compared with placebo (−60.0% m/s) and vitamin D2 (−52.0% m/s).

Consistent with our results, previous systematic reviews [16,18,23] that summarize the effect of oral vitamin supplement types on arterial stiffness show controversial results since there is no consensus among the findings in these systemic reviews. However, this NMA was designed to report the pooled results of the effects of different types of oral vitamin supplements on arterial stiffness and to provide evidence consistent with previous reviews.

Our findings show that vitamin D3 supplementation (by continuous intake for more than 12 weeks) is effective in reducing central arterial stiffness. This may be because endothelial cells present a vitamin D receptor and several enzymes capable of converting the circulating form of vitamin D (25-hydroxyvitamin) into the active form (1,25-dihydroxyvitamin D, ergocalciferol, or cholecalciferol) [56,57]. Thus, vitamin D2 and D3 can regulate endothelial and smooth muscle cell function by different mechanisms [58], with evidence showing a greater effect of vitamin D3 than vitamin D2, possibly because vitamin D3 maintains adequate vitamin D levels in the blood for a longer period of time [39,40]. Among the different functions modulated by vitamin D are antiproliferative effects on vascular smooth muscle, lymphocyte and monocyte differentiation, the release of proinflammatory cytokines [59], and modulation of the renin–angiotensin–aldosterone system [60]. These processes contribute to arterial stiffness and can induce monocyte infiltration into the vascular wall [61,62].

Looking at the results shown by the network geometry graph, we can see that there is ample evidence supporting the effectiveness of vitamin D3 in reducing central arterial stiffness. However, there are a small number of RCTs on other types of oral vitamin supplements, such as other forms of vitamin D and vitamins E, C, and B9, making it difficult to draw conclusions about whether these vitamins might also be effective. However, based on the findings of this study, it is evident that vitamins C and E could increase arterial stiffness (although not statistically significant), which would be conflicting with the current evidence [63,64,65]. Although previous studies support that vitamins C and E have no immediate effect on arterial stiffness, over time, they increase arterial stiffness [66,67]. Therefore, our results should be cautiously considered, until further RCTs are performed to clarify the direction of the effects of these vitamins.

This NMA has some limitations that should be acknowledged. First, the limited number of samples included in the assessment of some types of oral vitamin supplements and the likelihood that unpublished studies of these types of oral vitamin supplements might have changed the findings of the meta-analysis. Second, the findings of this NMA were derived after some manipulation of the data (ES of the raw data from the included studies), which might lead to some bias. Third, a wide variety of measurements of PWv (a-PWv, ba-PWv, br-PWv, cf.-PWv, and cr-PWv) were used, which may limit the implications of the results. Fourth, the overall risk of bias for RCTs showed some concerns of bias in most studies. Fifth, studies performed in different types of populations were included in the NMA (healthy, diabetes, hypertension, etc.); therefore, our results should be interpreted with caution. Sixth, because studies include different information on dose, frequency, and duration of treatment with oral vitamin supplements, it is not possible to estimate the threshold dose of vitamin D3 for PWv reduction. Finally, our findings are driven by the analysis of the effect of oral vitamin supplementation on PWv reduction, with consistent evidence supporting that PWv is a good predictor of CVD, CVD mortality, and all-cause mortality, with all the clinical and epidemiological consequences that this implies.

## 5. Conclusions

In conclusion, our study supports that oral vitamin D3 supplementation for more than 12 weeks could be an effective approach to reduce central arterial stiffness. Our findings are based on data from experimental studies and provide the best currently existing evidence of the effects of oral vitamin supplementation on arterial stiffness as a therapeutic and preventive strategy. However, future well-designed and statistically powered RCTs are essential to reinforce the currently limited evidence to reflect that, over time, patients at high risk of CVD could benefit from the effects of oral vitamin supplementation, specifically vitamin D3, as a useful strategy to improve their vascular health.

## Figures and Tables

**Figure 1 nutrients-14-01009-f001:**
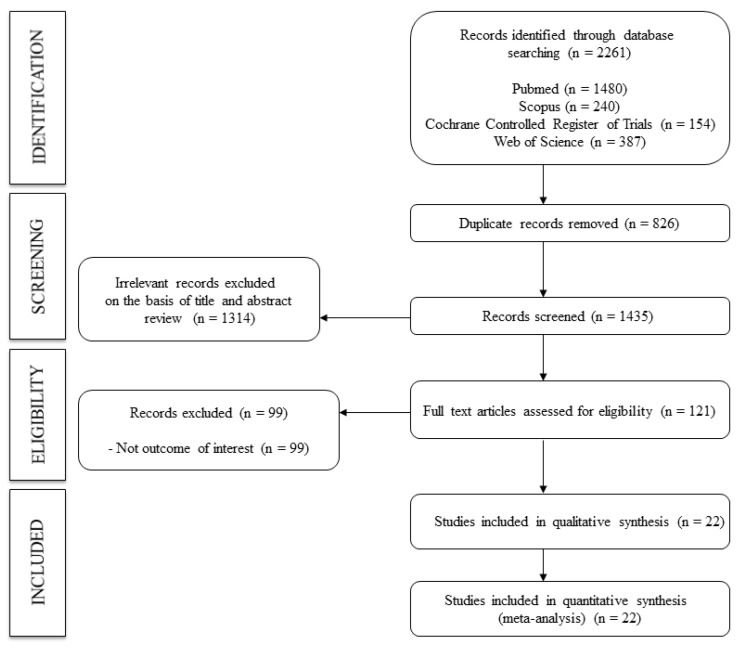
Flowchart.

**Figure 2 nutrients-14-01009-f002:**
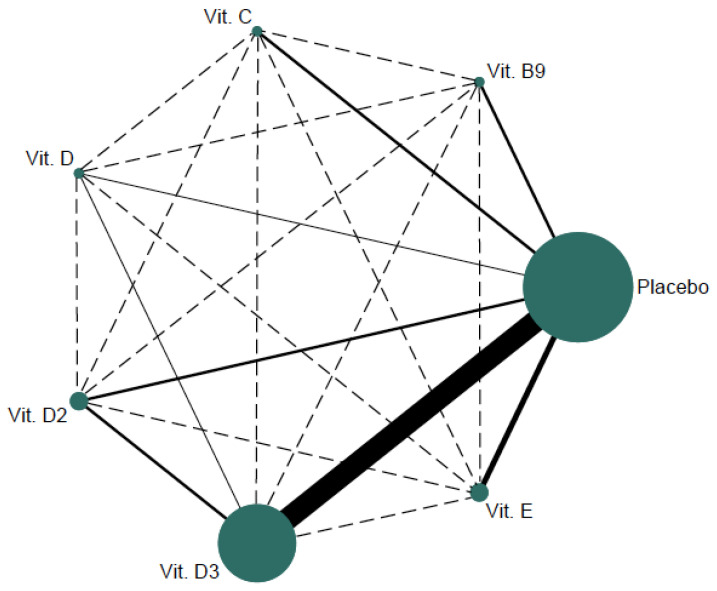
Network of available comparisons between different types of oral vitamin supplements in arterial stiffness.

**Table 2 nutrients-14-01009-t002:** Pooled mean differences in different types of oral vitamin supplements on arterial stiffness.

Placebo	−0.14 (−0.69, 0.42)	0.17 (−0.29, 0.63)	−0.04 (−0.56, 0.47)	−0.24 (−0.50, 0.01)	−0.08 (−0.24, 0.08)	0.20 (−0.17, 0.58)
−0.13 (−1.02, 0.77)	Folic acid (vit. B9)	NA	NA	NA	NA	NA
0.36 (−0.52, 1.24)	0.49 (−0.77, 1.74)	Ascorbic acid (vit. C)	NA	NA	NA	NA
−0.01 (−0.63, 0.61)	0.12 (−0.97, 1.21)	−0.37 (−1.44, 0.71)	Calcitriol (vit. D)	NA	−0.32 (−0.84, 0.20)	NA
−0.10 (−0.73, 0.54)	0.03 (−1.07, 1.13)	−0.45 (−1.54, 0.63)	−0.09 (−0.96, 0.79)	Ergocalciferol (vit. D2)	**−0.25** **(−0.48, −0.02)**	NA
−0.21 (−0.54, 0.11)	−0.09 (−1.04, 0.86)	−0.57 (−1.51, 0.36)	−0.21 (−0.87, 0.45)	−0.12 (−0.76, 0.52)	Cholecalciferol (vit. D3)	NA
0.28 (−0.36, 0.92)	0.41 (−0.69, 1.51)	−0.08 (−1.17, 1.01)	0.29 (−0.60, 1.18)	0.38 (−0.53, 1.28)	0.50 (−0.22, 1.22)	Tocotrienol (vit. E)

NA: not available.

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
