# Peer review of "The Comparative Effects of Different Types of Oral Vitamin Supplements on Arterial Stiffness: A Network Meta-Analysis"

_nutrients, 2022, doi:10.3390/nu14051009_

Round 1

Reviewer 1 Report

Summary of manuscript: This systematic review aimed to assess the effects of vitamin supplements on arterial stiffness in human randomized controlled trials. The results demonstrated that the different oral vitamin supplements did not impact arterial stiffness; however, vitamin D3 reduced arterial stiffness compared to placebo and vitamin D2. It was concluded that oral vitamin D3 supplementation for more than 12 weeks reduces arterial stiffness and could be used to improve vascular health.

General comments: I carefully reviewed this manuscript. The authors provided a meticulous systematic review. The main concerns are formatting issues, which can easily be fixed. In addition, it would be helpful if the Introduction and Discussion sections included clarifying information. I provided my specific comments below.    

Introduction

Point 1: Line 33: I suggest removing “The” and starting the sentence with “Arterial”

Point 2: Line 54: Should “51” be present in this sentence?

Point 3: Specifically, why were folic acid, vitamin C, vitamin D, and vitamin E selected for this study? Was it simply due to the lack of published articles in this area? Are these vitamins associated with proposed mechanisms regarding arterial stiffness? It would be helpful to clarify this issue in the Introduction.

Materials and Methods

Point 4: Line 78: Should “74” be present in this sentence?    

Point 5: Line 134: Should “122” be present in this sentence?  

Point 6: Line 158: Should “147” be present in this sentence? I am not certain what these numbers are referring to throughout the manuscript. Please review the manuscript for these numbers.  

Results

Point 7: Line 205: Can you please provide the definition for ES?

Point 8: Figure 1: In the screening category, the number is missing for the irrelevant records.

Point 9: Line 215: Should this sentence state “between different types”?

Point 10: Table 2: In the table legend, it would be nice to include the definition of NA.

Point 11: Line 240: Please check the number “219”

Discussion

Point 12: Lines 260-269: The authors did a good job describing the mechanisms for vitamin D3; however, it would be helpful to discuss the differences between vitamin D2 and vitamin D3. Is vitamin D2 also converted to the active form of vitamin D? A few sentences should be sufficient to clarify these differences.  

Point 13: Line 261: “tan” should be “than”

Point 14: Line 266: Please check the number “243”

Point 15: Line 292: Please check the number “268”

Author Response

Reviewer 1

Summary of manuscript: This systematic review aimed to assess the effects of vitamin supplements on arterial stiffness in human randomized controlled trials. The results demonstrated that the different oral vitamin supplements did not impact arterial stiffness; however, vitamin D3 reduced arterial stiffness compared to placebo and vitamin D2. It was concluded that oral vitamin D3 supplementation for more than 12 weeks reduces arterial stiffness and could be used to improve vascular health.

General comments: I carefully reviewed this manuscript. The authors provided a meticulous systematic review. The main concerns are formatting issues, which can easily be fixed. In addition, it would be helpful if the Introduction and Discussion sections included clarifying information. I provided my specific comments below.    

Authors:

Thank you for the reviewer´s comment. We greatly appreciate the time the reviewer spent reviewing the manuscript.

Introduction

Point 1: Line 33: I suggest removing “The” and starting the sentence with “Arterial”

Authors:

Thank you for the reviewer's comment. As suggested, we have modified this sentence, as follows:

Arterial stiffness is related to the onset of vascular ageing [1] and could be considered a contributing factor for the development of cardiovascular disease (CVD) [2].”

Point 2: Line 54: Should “51” be present in this sentence?

Authors:

Thank you for the reviewer's comment. As suggested, we have removed this mistake.

Point 3: Specifically, why were folic acid, vitamin C, vitamin D, and vitamin E selected for this study? Was it simply due to the lack of published articles in this area? Are these vitamins associated with proposed mechanisms regarding arterial stiffness? It would be helpful to clarify this issue in the Introduction.

Authors:

Thank you for the reviewer's comment. As suggested, we have clarified this issue, as we have included all available evidence on the effect of vitamin supplements on arterial stiffness.

In the introduction section:

“Although there are several previously published systematic reviews and meta-analyses [16-23] on the effects of different types of oral vitamin supplements on arterial stiffness and/or endothelial function, and, although the number of samples included in the assessment of some types of oral vitamin supplements is limited (and not all types of vitamins have been tested for arterial stiffness), none of them have quantitatively assessed the effects of different types of oral vitamin supplementation (folic acid or vitamin B9, ascorbic acid or vitamin C, calcitriol or vitamin D, ergocalciferol or vitamin D2, cholecalciferol or vitamin D3, tocotrienol or vitamin E) on arterial stiffness. […]”

Materials and Methods

Point 4: Line 78: Should “74” be present in this sentence?    

Authors:

Thank you for the reviewer's comment. As suggested, we have removed this mistake.

Point 5: Line 134: Should “122” be present in this sentence?  

Authors:

Thank you for the reviewer's comment. As suggested, we have removed this mistake.

Point 6: Line 158: Should “147” be present in this sentence? I am not certain what these numbers are referring to throughout the manuscript. Please review the manuscript for these numbers.  

Authors:

Thank you for the reviewer's comment. As suggested, we have removed this mistake.

Results

Point 7: Line 205: Can you please provide the definition for ES?

Authors:

The reviewer´s comment seems judicious. As suggested, we have included the definition for ES in the methods section, as follows:

“[…] For this purpose, we used the DerSimonian–Laird random effects method [29] to calculate a pooled effect size (ES) estimate and the respective 95% confidence intervals (CI), and we estimated the pooled percentage change of m/s for oral vitamin supplement interventions. […].”

Point 8: Figure 1: In the screening category, the number is missing for the irrelevant records.

Authors:

Thank you for the reviewer's comment. As suggested, we have included the missing number for irrelevant records in figure 1.

Point 9: Line 215: Should this sentence state “between different types”?

Authors:

The reviewer´s comment seems judicious. As suggested, we have modified this sentence, as follows:

Figure 2. Network of available comparisons between different types of oral vitamin supplements in arterial stiffness.”

Point 10: Table 2: In the table legend, it would be nice to include the definition of NA.

Authors:

The reviewer´s comment seems judicious. As suggested, we have included the definition of NA in the legend of table 2.

“NA: not available.”

Point 11: Line 240: Please check the number “219”

Authors:

Thank you for the reviewer's comment. As suggested, we have removed this mistake.

Discussion

Point 12: Lines 260-269: The authors did a good job describing the mechanisms for vitamin D3; however, it would be helpful to discuss the differences between vitamin D2 and vitamin D3. Is vitamin D2 also converted to the active form of vitamin D? A few sentences should be sufficient to clarify these differences.  

Authors:

The reviewer´s comment seems judicious. As suggested, we have clarified this issue in the discussion section.

“[…] This may be because endothelial cells present a vitamin D receptor and several enzymes capable of converting the circulating form of vitamin D (25-hydroxyvitamin) into the active form (1,25-dihydroxyvitamin D, ergocalciferol or cholecalciferol) [56, 57]. Thus, vitamin D2 and D3 can regulate endothelial and smooth muscle cell function by different mechanisms [58], with evidence showing a greater effect of vitamin D3 compared to vitamin D2, possibly because vitamin D3 maintains adequate vitamin D levels in the blood for a longer period of time [39,40]. […]”

Point 13: Line 261: “tan” should be “than”

Authors:

The reviewer´s comment seems judicious. As suggested, we have replaced “tan” to “than”, as follows:

“Our findings show that vitamin D3 supplementation (by continuous intake for more than 12 weeks) is effective in reducing arterial stiffness. […]”

Point 14: Line 266: Please check the number “243”

Authors:

Thank you for the reviewer's comment. As suggested, we have removed this mistake.

Point 15: Line 292: Please check the number “268”

Authors:

Thank you for the reviewer's comment. As suggested, we have removed this mistake.

Reviewer 2 Report

This is a well-written meta-analysis on the effect of different type of vitamin supplements on arterial stiffness, with significant clinical implications.

However, several issues should be addressed:

  1. Authors conclude that the effect of vitamin D3 on arterial stiffness is significant after 12 weeks of intake. Which arterial stiffness? It is not clear in the main text. According to the Table S7, central stiffness is reduced, whereas peripheral stiffness is increased. Moreover, in Table S7, the reduction in central stiffness is referred to vit. B9 compared to placebo and not in vit.D compared to placebo. Would it be more appropriate to present data on vit. D3 and D2 separately instead of vit D in Table S7? Given that central (and not peripheral) arterial stiffness is a strong predictor of CV risk, it is the effect of vit D3 on central arterial stiffness that should be emphasized on the discussion.
  2. The studies that are included in the meta-analysis consist of different type of populations. Whether the benefit of D3 on central arterial stiffness includes all type of populations or the magnitude of this effect is different among healthy adults, patients with DM and CHF, can not be demonstrated by the present results. A comment on the Discussion should be added.
  3. Moreover, given the different dose of vit.D that is used in included studies, could the threshold dose of vit D3 for the reduction of PWV be estimated? It is another issue that needs discussion. 

Minor changes:

  1. Abstract, 1st line: The phrase 'Arterial stiffness, a subclinical process that contributes to the development of cardiovascular disease..' should better change to: Arterial stiffness, a significant prognostic factor of cardiovascular disease, may be affected...
  2. Introduction, line 3: possible causes of arterial stiffness. Replace 'causes' to 'mechanisms'.
  3. Introduction, line 8: PWV is the reference predictor. Replace with the phrase 'PWV measurement is the gold standard method for the non-invasively assessment of arterial stiffness.
  4. Discussion, last 3 lines: 'PWV (add V) is a good predictor of CVD, it is indeed a surrogate measure for the main outcomes appropriate 268 for these interventions...' Please, rephrase. It does not make sense.
  5. Several numbers (like 268 in the above sentence) are added throughout the text. Type-errors?

Author Response

Reviewer 2

This is a well-written meta-analysis on the effect of different type of vitamin supplements on arterial stiffness, with significant clinical implications.

Authors:

Thank you for the reviewer´s comment. We greatly appreciate the time the reviewer spent reviewing the manuscript.

However, several issues should be addressed:

  1. Authors conclude that the effect of vitamin D3 on arterial stiffness is significant after 12 weeks of intake. Which arterial stiffness? It is not clear in the main text. According to the Table S7, central stiffness is reduced, whereas peripheral stiffness is increased. Moreover, in Table S7, the reduction in central stiffness is referred to vit. B9 compared to placebo and not in vit.D compared to placebo. Would it be more appropriate to present data on vit. D3 and D2 separately instead of vit D in Table S7? Given that central (and not peripheral) arterial stiffness is a strong predictor of CV risk, it is the effect of vit D3 on central arterial stiffness that should be emphasized on the discussion.

Authors:

The reviewer´s comment seems judicious. As suggested, we have clarified this issue in the discussion and conclusion sections.

  1. The studies that are included in the meta-analysis consist of different type of populations. Whether the benefit of D3 on central arterial stiffness includes all type of populations or the magnitude of this effect is different among healthy adults, patients with DM and CHF, can not be demonstrated by the present results. A comment on the Discussion should be added.

Authors:

The reviewer´s comment seems judicious. As suggested, we have included this issue in the discussion section.

“[…] Fifth, studies performed  in different types of populations were included in the NMA (healthy, diabetes, hypertension,…); therefore, our results should be interpreted with caution. […]”

  1. Moreover, given the different dose of vit.D that is used in included studies, could the threshold dose of vit D3 for the reduction of PWV be estimated? It is another issue that needs discussion. 

Authors:

The reviewer´s comment seems judicious. As suggested, we have included this issue in the discussion section.

“[…] Sixth, because studies include different information on dose, frequency and duration of treatment with oral vitamin supplements, it is not possible to estimate the threshold dose of vitamin D3 for PWv reduction. […]”

Minor changes:

  1. Abstract, 1st line: The phrase 'Arterial stiffness, a subclinical process that contributes to the development of cardiovascular disease..' should better change to: Arterial stiffness, a significant prognostic factor of cardiovascular disease, may be affected...

Authors:

Thank you for the reviewer's comment. As suggested, we have modified this sentence, as follows:

“Arterial stiffness, a significant prognostic factor of cardiovascular disease, may be affected by dietary factors. […]”

  1. Introduction, line 3: possible causes of arterial stiffness. Replace 'causes' to 'mechanisms'.

Authors:

Thank you for the reviewer's comment. As suggested, we have replaced 'causes' to 'mechanisms', as follows:

“[…] Inflammation and oxidative stress are possible mechanisms of arterial stiffness. […]”

  1. Introduction, line 8: PWV is the reference predictor. Replace with the phrase 'PWV measurement is the gold standard method for the non-invasively assessment of arterial stiffness.

Authors:

Thank you for the reviewer's comment. As suggested, we have modified this sentence, as follows:

“[…] Pulse wave velocity (PWv) measurement is the gold standard method for the non-invasively assessment of arterial stiffness [2,6]. […]”

  1. Discussion, last 3 lines: 'PWV (add V) is a good predictor of CVD, it is indeed a surrogate measure for the main outcomes appropriate 268 for these interventions...' Please, rephrase. It does not make sense.

Authors:

The reviewer´s comment seems judicious. As suggested, we have modified this sentence, as follows:

“[…] Finally, our findings are driven by the analysis of the effect of oral vitamin supplementation on PWv reduction, with consistent evidence supporting that PWv is a good predictor of CVD, CVD mortality and all-cause mortality, with all the clinical and epidemiological consequences that this implies.

  1. Several numbers (like 268 in the above sentence) are added throughout the text. Type-errors?

Authors:

Thank you for the reviewer's comment. As suggested, we have removed these mistakes in the manuscript.
